# Identifying pandemic-related stress factors from social-media posts - effects on students and young-adults

**Sachin Thukral,**\* **Suyash Sangwan,**† **Arnab Chatterjee,**‡ **Lipika Dey**§ (*TCS Research, India*)

## Abstract

The COVID-19 pandemic has thrown natural life out of gear across the globe. Strict measures are deployed to curb the spread of the virus that is causing it, and the most effective of them have been social isolation. This has led to wide-spread gloom and depression across society but more so among the young and the elderly. There are currently more than 200 million college students in 186 countries worldwide, affected due to the pandemic. The mode of education has changed suddenly, with the rapid adaptation of e-learning, whereby teaching is undertaken remotely and on digital platforms. This study presents insights gathered from social media posts that were posted by students and young adults during the COVID times. Using statistical and NLP techniques, we analyzed the behavioral issues reported by users themselves in their posts in depression-related communities on Reddit. We present methodologies to systematically analyze content using linguistic techniques to find out the stress-inducing factors. Online education, losing jobs, isolation from friends, and abusive families emerge as key stress factors.

## 1 Introduction

The outbreak of novel Coronavirus disease 2019 (COVID-19) has affected the day to day life of people across the world and is slowing down the global economy (World Health Organization, 2020). Along with serious health problems leading to an alarming number of fatalities over a very short time, the world is also seeing a staggering rise in COVID-19 linked depression, anxiety, violence, and suicidal tendencies across different sections of

---
\*sachi.2@tcs.com
†suyash.sangwan@tcs.com
‡arnab.chatterjee4@tcs.com
§lipika.dey@tcs.com

the society. Our study of a few communities in the Reddit platform which discuss depression related issues was initiated to analyze social media content related to pandemic-induced stress markers. We chose this platform since a large number of people who suffer from depression and/or related symptoms express themselves freely here. However, during analysis, we encountered an alarmingly large number of posts that discussed student-related issues. Hence we narrowed down our study to focus primarily on this community. Our study revealed that this community contained high school students, college-goers as well as young adults on the verge of graduating from college, most of whom suffer from mental health issues. We intend to apply Natural Language Processing (NLP) and text mining techniques to analyze such content and generate insights about the different types of concerns expressed by student members of these communities.

Several surveys have also reported a phenomenal rise in the number of depression-related cases among college students between the end of March through May 2020. A research project based at the University of Michigan, Boston University, and the University of California, Los Angeles have reported over a 5% rise in depression cases when compared with the data of last fall. Suicide risks are also reported to have gone up by over 2% as per a survey taken in April by Active Minds—which has chapters in more than 500 colleges in the USA (Petersen, Andrea, 2020). Another recent, survey-based study on university students in Bangladesh (Islam et al., 2020) revealed that students experienced pronounced depression and anxiety, with around 15% reportedly had moderately severe depression, while 18.1% were severely suffering from anxiety.

Our work differs substantially from the above since we analyze social media content and not surveys or feed-backs. Social media data is informal

and often much more expressive, giving us cues about possible causes along with the effects. An important aspect of social media is that it is interactive in nature – users can reply, discuss, debate, and possibly agree or disagree on opinions, which bring out rich perspective to their points of view, which is not possible in a survey that usually collects *one-shot* answers to queries. With social isolation and distancing from physical friends, social media also saw a huge rise in footfalls during the pandemic.

In this paper, we have presented a detailed analysis of depression-related posts from specific Reddit communities posted during the COVID-19 period. Starting with the community at large, we systematically identified content related to students or fresh graduates who have reported depression or mental health problems suffer and derived some novel insights about COVID-induced stress factors. The key contributions of the paper are as follows :

1. Identified distinct statistically valid linguistic markers that differentiate depression-related content from other general content. This is done with the help of established linguistic analysis tools like LIWC.

2. Propose a text classification method to identify student-authored or student-related content from within a large repository

3. Proposed topical analysis of COVID-19 time posts using LDA and methods to perform subsequent linguistic analysis of representative posts for each topic.

4. Provided detailed insights on pandemic induced stress in students – A set of detailed insights are presented through a topical analysis of content. While some of these issues were found to be quite generic and persistent, it was observed that the pandemic added more stress factors to people, specifically students and young adults, who were already suffering from mental health issues.

5. The methods proposed above can be generalized to identify the right social media content for analytical purposes and generate insights systematically.

**About the Data source:** We have chosen **Reddit**, which is an American content aggregation platform, with more than 330 million users, called *Redditors*, as our content platform. Contents on this site are divided into categories or communities known as *subreddits*. More than $138,000$ of these active communities share, debate and discuss various issues of public interest. Due to its large user base and high levels of user engagement, Reddit is considered as a good source for gathering public insights on any issue of current interest. Reddit is known to be a widely used online forum and social media site among the college students (Duggan and Smith, 2013). Our analysis is based on content that was posted between December 2019 and June 2020. More details about the dataset are given in Section 3.

Section 2 presents an overview of related work on analyzing social media content for detecting the mental health of people. Section 4 presents details about the analytical methods used for our analysis. Section 5 presents the key insights that are found to affect students and young adults during COVID 19.

## 2 Related Work

Since the outbreak of this pandemic, researchers have been extensively studying its impact on the mental health of people. One of the studies, where researchers have examined the sentiment dynamics of people (Zhou et al., 2020), showed that various government policies and the events did affect people's overall sentiment from positive to negative. Researchers also studied that whether engaging with online community support initiatives (OCSIs) generated a positive or negative mood in those who engaged with them and whether or not different reasons for this engagement play different roles (Elphick et al., 2020). They showed that people were in a better mood after engaging with an OCSI than before. Another study conducted on Twitter discussions, identified popular unigrams, bigrams, salient topics and themes, and sentiments in collocated tweets (Xue et al., 2020). Another study about the impact of COVID-19 on students with disabilities or health concerns (Zhang et al., 2020) showed that people with disabilities and health concerns are more worried about classes going online than their peers without disabilities. They argued that these people need more confidence in the accessibility of the online learning tools that are becoming more prevalent in these times. A study using Google trends to see the impact of lockdown on the mental health of people (Brodeur et al., 2020) has shown that during lockdown there is a significant increase

in google searches on boredom, loneliness, worry, and sadness. Another study focuses on the things that are we depressed about during COVID-19 (Li et al., 2020). They trained deep models that classify tweets into emotions like anger, anticipation, disgust, fear, joy, sadness, surprise, and trust, and have also built an EmoCT dataset for classifying these tweets. They also applied methods to understand why the public may feel sad or fearful during these times. There is one study that highlights differences in topics concerning emotional responses and worries based on gender (van der Vegt and Kleinberg, 2020). They used Real World Worry Dataset (RWWD) which is a text data collected from 2500 people in which each participant was asked to report their emotions and worries (Kleinberg et al., 2020). They reported that women worried mainly about loved ones and expressed negative emotions such as anxiety and fear whereas men were more concerned with the societal impacts of the virus. A study conducted using Reddit platform (Murray et al., 2020) used *r/COVID19Positive* subreddit in which patients who are COVID-positive talk about their experiences and struggles with the virus. While providing a perspective of the patient experience during there course of treatment, they also suggest that there is a need for other supports such as mental health support during these times. Our work is focused on studying the effects of the pandemic on students who suffer from mental health issues.

## 3 Creating a focused dataset for the study

Since the Reddit platform is well organized around communities or subreddits with a shared interest, our first objective was to identify proper communities that contained data appropriate for mental health analysis. Based on the community descriptions, we found 47 subreddits whose members discuss mental health-related issues. However, many of them are not active. Further, some of these groups focused on specific neurotic disorders like autism or Bi-polar disease, etc. These were also not considered for our study. Finally, we have considered only those posts from the depression and mental health subreddits were combined to create a *focused* data set. We used the *Pushshift API* (pushshift.io, 2020), provided by the platform, to crawl the data for the period from July 2019 to April 2020. Detailed statistics about the datasets

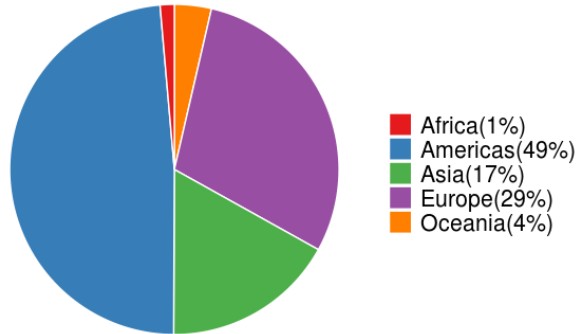

Figure 1: Geographical spread of posts

are presented in the Table 1.

The posts themselves do not have any geotagging. However, to assess the geographical spread, we looked for location names referred to in the content itself. There are 7% of posts that have mentions of location in our data. After doing appropriate resolution of the location names, identified using the Spacy Named Entity Recognition tool, we observed that the data predominantly refers to the Americas and Europe, with 49% and 29% representation respectively(Figure 1). 17% refer to Asian countries and the rest are from other parts of the world.

## 4 Finding Language markers for Depression-related text content

To determine the distinct markers, if any, for depression, we did the following. We obtained data from two popular subreddits *r/AskReddit* and *r/worldnews*, where large numbers of users engage in discussing News and other contemporary topics, for the same time period to create a *control set*. We did a numerical and linguistic comparison of this data with the depression-related posts in the focused set. To compare the statistics, we divided the time-period into two phases - the pre-COVID phase ranging from July to November 2019 while the other phase lasted from December 2019 to April 2020. Table 1 shows that these two sets appear to be very similar in nature when numbers are compared. However, the linguistic analysis revealed something very different.

For the linguistic comparison of the two sets of posts, we analyzed the relative usage of different categories words in the two data sets for the entire periods. We have combined the posts from the control set $\mathcal{C}_1$ and control set $\mathcal{C}_2$ and compared with the combined posts from focused set $\mathcal{F}_1$ and focused set $\mathcal{F}_2$. This is done with the help of a linguis-

Table 1: Dataset description

| | July-November 2019 | |
|---|---|---|
| | control set $\mathcal{C}_1$ | focused set $\mathcal{F}_1$ |
| Total posts | 2,412,600 | 135,078 |
| Total User | 679,099 | 79,691 |
| Comments | 41,607,923 | 465,286 |
| | December 2019-April 2020 | |
| | control set $\mathcal{C}_2$ | focused set $\mathcal{F}_2$ |
| Total posts | 2,661,403 | 139,888 |
| Total User | 722,520 | 81,063 |
| Comments | 42,522,951 | 454,145 |

tic tool called Linguistic Inquiry and Word Count (LIWC). It uses a dictionary created by psychological experts for categorization of words (Tausczik and Pennebaker, 2010) into 64 categories that fall under different groups like psychological processes, personal concerns, social processes, linguistic properties, etc. LIWC categories were created for and by the psychologists to assess various attributes like attentional focus, emotionality, social relationships, thinking styles, and individual differences of authors, based on text content created by them.

Using the LIWC tool, we transform each post into a 64-dimensional vector that represents its LIWC category concentrations. Using these category concentrations as features, we performed the two-sample Kolmogorov-Smirnov (KS) test to find out the distinguishing features, if any, from the two sets. The two-sample K–S test is one of the most useful and general nonparametric methods for comparing two samples since it is sensitive to differences in both location and shape of the empirical cumulative distribution functions of the two samples. The null hypothesis for the KS test is that both groups were sampled from populations with identical distributions. The KS distance metric $D$ is the maximum distance between the Cumulative Distributive Functions of the corresponding features from both the sets.

For each of 64 LIWC category we have computed cumulative distribution in each set. Denoting the cumulative distribution of focused set for $k$-th LIWC category as $F_{1,n}^k(x)$ and cumulative distribution for the control set for $k$-th LIWC category with $F_{2,m}^k(x)$ where $n$ and $m$ are the cardinalities for the two sets respectively. Then the KS distance metric $D$ is computed as $D_{n,m} = sup_x|F_{1,n}^k(x) - F_{2,m}^k(x)|$. For the null hypothesis

to be rejected $D_{n,m} > c(\alpha)\sqrt{\frac{n+m}{n.m}}$. We rejected the null hypothesis at level $\alpha = 0.001$.

Our analysis revealed that the two sets significantly differed in word usage for 32 of the 64 categories. The most significant differences are summarized as follows.

1. *Negative emotion* category words like *hurt*, *ugly*, *nasty*, etc. occur far more often in the focused set containing depression related posts than in the control set.

2. The focused set also has higher usage of *Affect* category words like *cried, trauma, victim*, etc.

3. Use of words like *pain*, *sick*, *heal*, etc. that belong to the *health* category are also significantly more frequent in this set.

4. The control set shows a much higher use of words like *mate*, *society* etc. which fall under the *social* category.

5. It is observed that, the focused set shows a significantly higher use of first person singular words like *I*, *me* and *myself* over the control set. On the contrary, the control set has a much higher use of the second person singular words like *you*, *your* and *you'll*. Literature on psychological studies of text for personality assessment (Majumder et al., 2017), (Pennebaker et al., 2001) have pointed out that use of pronouns can be a significant indicator of personalities.

The above findings indicate that there are clear linguistic markers to differentiate depression-related posts from the general talk. The posts in the focused set are mostly about stress, health-related concerns, negative thoughts shared by people, which are known indicators of depression. This reinforces the fact that the chosen subreddits indeed have the right content to gather depression-related information. In the next section, we present how pandemic specific issues were identified by comparing the content from the two different phases.

## 5 Finding pandemic-induced stress factors

Our next intent was to find distinct markers of stress, if any, during the COVID period. We separated the focused set data into two subsets - *pre-COVID* set ($\mathcal{F}_1$) which contained content from July

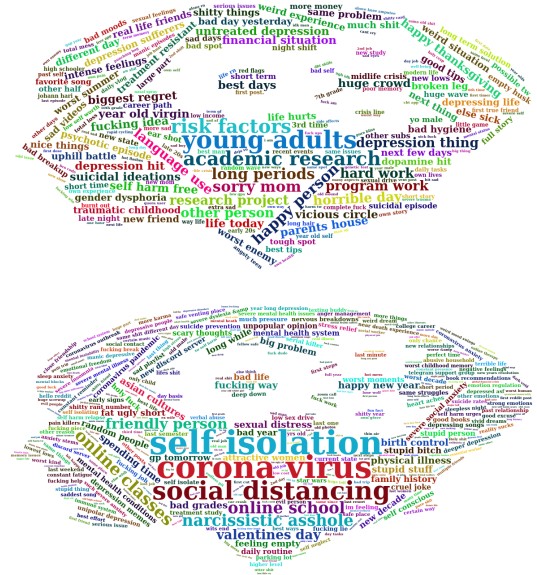

Figure 2: The keywords extracted from the depression set in two time periods – pre-COVID (Top) and COVID (Bottom).

to November, 2019 and *COVID*-time ($\mathcal{F}_2$) content that spanned over December 2019 to April 2020.

In order to do the comparison, first the Rapid Automatic Keyword Extraction (RAKE) algorithm (Rose et al., 2010) was used to extract the significant terms from each post and thereafter use these terms to represent the content. RAKE is a well known key term extraction method that uses statistical analysis of words and phrases in the collection to determine the key-terms. After removing the stop words like *a, an, the*, etc. from the content, it detects phrase delimiters in each post. Using a co-occurrence matrix of words built from the entire repository, it then assigns a score to each pair based on its relative co-occurrence frequency in the collection. The algorithm then creates a matrix of word co-occurrences and assigns a score to each word as a function of its individual occurrence and co-occurrence with other words. This is continued for all words and phrases. Higher the score of a term, higher is its representation in the corpus.

Figure 2 shows the tag clouds generated by the key-terms of each sub-set. The lower tag-cloud clearly shows that terms like *social distancing*, *self isolation*, *corona virus*, etc. had significant presence in the COVID-time content. Since these terms occurred in conjunction with the earlier mentioned LIWC categories like *health, affect*, etc. these can be considered as the stress inducers for this period.

A significant aspect that we noticed in the

COVID-time content, was the presence of the terms *online education*, *online school*, *online classes*, *assignments*, etc. Thus, we decided to dig deeper into this aspect. However, rather than only looking at online-education related content, we extended our analysis towards analyzing content that could be discussing issues about students, in general. This is discussed in detail in the next sub-section.

## 5.1 Detection and Analysis of content related to Student Community

For this study we are considering the content from the focused set $\mathcal{F}_2$. Since there is no meta-data available to identify posts that are by, for or about students, we employed a classifier to identify relevant content for the study. We started by manually annotating a few posts that were picked up using a set of student-specific key-terms like *online education, assignments, grades, class, lecture*, etc. A total of $1,100$ posts from the COVID period were annotated as positive samples discussing student related issues. An equal number of negative samples were picked up from the remaining data set, ensuring that they did not contain any student-specific issue.

After trying out a range of classifiers with different features, the Extreme Gradient Boosting classifier (Friedman, 2001) using word level TF-IDF as features was found to achieve the highest ten-fold cross-validation accuracy of $96.19\%$ on the annotated data set. This, when applied on the remaining posts of COVID times, identified $19,552$ posts are related to students. This number constitutes around $14\%$ of the posts which is significantly high. It was not possible to manually verify each of these posts, however a random check revealed that the selected content were indeed positive samples. Since our intent was not to generate exact statistics, but rather identify indicative issues from social media, this should be acceptable.

These $19,552$ posts were then subjected to topic extraction using unsupervised Latent Dirichlet Allocation (LDA). LDA is a fairly standard generative model that works as follows – it assumes that a document was generated by picking a set of topics and then for each topic picking up a set of words. Starting with an input $k$, which specifies the number of topics, the output of LDA contains the distribution of these $k$ topics in each document. It also provides the distribution of words in each topic. For the present content, the best result with minimum

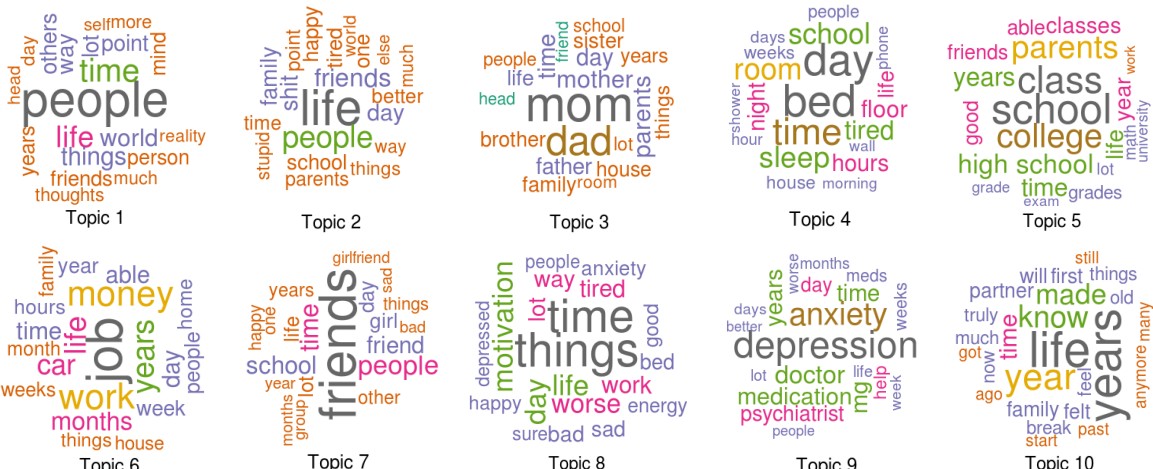

Figure 3: Word clouds for the topics extracted by the LDA – Topic 1: People and social issues, Topic 2: Negative emotions, Topic 3: Family, Topic 4: Daily activities, Topic 5: Educational concerns, Topic 6: Employment concerns, Topic 7: Friendship, Topic 8: Feelings, Topic 9: Health concerns, Topic 10: Past issues.

perturbation was obtained for $k$ equal to 10.

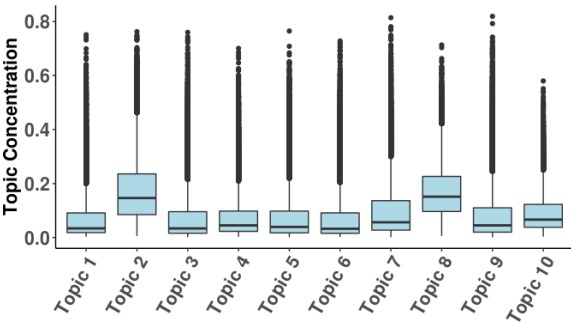

Figure 4: Boxplot of topic distribution in posts.

Figure 3 presents overview of the ten topics. It can be seen that the significant terms of the topics are unique. We used these to manually associate indicative names to the topics. The boxplots presented in Figure 4 clearly show that all the topics are positively skewed, i.e. there are a few outliers with very high score for each topic. Topics 2 and 8, which indicate negative emotions and feelings have a steady underlying presence in almost all the documents. All other topics have quite a few outliers with very high scores. Of course, as per LDA norms, each post is a mix of various topics. In the next subsection, we present more detailed discussions around our findings.

## 5.2 Key Insights on pandemic - induced additional stress for the young adults who suffer from depression

We now present detailed insights on some of the key topics individually.

### 5.2.1 Online Education related stress

In this section, we provide the insights gathered from the posts related to the topic of online education. These posts on an average contained 200 to 300 words, which is enough to explain issues in detail. The group consists of students belonging to high school and college both, from all over the world, with more users from the earlier group. Table 2 presents the key aspects identified as stress inducers resulting from online classes, along with portions from sample posts.

Along with "heavier workload than regular times" - noticeable is the presence of "fear of missing assignments". This was found to be a predominant fear and cause of mental anguish in this group, which also appears to work in a self-feeding loop. More the fear, more the apathy to work and thus more accumulated distress.

Factors like "missing friends" and "missing counsellors" are very crucial issues. Not being able to meet friends and counsellors physically leads to increased mental stress. The posts show that friends and counsellors play an important and positive role in their well-being during normal school or college times. There are also a few counsellors within this group offering positive advice to students. Conversations within communities and their influences, can be subject of a separate study.

The other aspect that emerged was the mention of "parents" playing an abusive role. It appears that several students who suffer from depression felt even more tortured by the isolation since they had to deal with non-cooperative and abusive parents at

home. The posts reveal that a number of students find it difficult to face parents who are not very supportive of their "depression patient" status.

**Discussions** - The fact that online classes do tend to stretch beyond usual hours is a major stress-inducing factor and is undesirable. Discipline similar to regular physical classes should be maintained. Various studies have reported the supportive roles of teachers and classmates in physical schools towards people suffering from emotional distress. Our study reinforces this. While online education is being considered on a wider scale worldwide, often as a substitute for regular classroom teaching in the near future, then there is an immediate need to build new mechanisms for introducing checks and balances to ensure inclusivity. It is important to embed it in a holistic environment that can take track of the mental health of the students and ensure their well-being. While information can be disseminated through technology, it is difficult to substitute the role played by the physical presence of other humans in a student's life, particularly for those who lack much family support. However, this aspect has to be integrated into a learning environment.

### 5.2.2 Sentiment analysis for people mentions

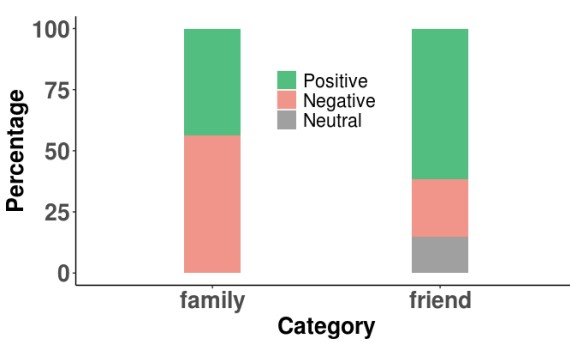

Figure 5: Sentiment distribution in posts related to friend and family.

Continuing with the discussion in the earlier section, we provide more details about two different contexts in which people mention *friends* and *parents* are found in these posts in general. We did a sentiment analysis of all posts containing high scores for topics 7 and 3 indicative of *friends* and *parents* respectively. Figure 5 presents the relative occurrence of positive and negative sentiments for these two relations. While the term *friend* occurs more with a positive connotation, terms like *mom, dad, parents* occurred more with negative connotations. These are consistent with the observations

made in section 5.2.1. The reasons are also pretty similar. Friends are *missed* and mentioned as important part of *support systems*. On the other hand *parents* are most often mentioned as *abusive*. Many have reported *history of abuse from childhood* and also family history of mental troubles. Table 3 shows portions of sample text picked up from posts scoring high on topic 3 and topic 7. Though the parental issue cannot be attributed to the pandemic as such, social isolation, quarantine and lockdown forced the students to spend more time at home and with parents - this can be considered as a severe issue of the times.

### 5.2.3 Insights related to Employment concerns

We now discuss a few insights gathered from a different category of students who had recently graduated or completed high school. This group of people is majorly found to talk about "losing job". Job losses have been reported widely in the media. These posts especially reveal the trauma of the group which consists of young adults who suffer from depression and find a job with a lot of difficulties. Quite obviously, losing a new job along with financial burden appears to be immensely stressful for them. A second stress factor that is found in the COVID-19 induced isolation at a new job location, which makes it difficult for them to maintain healthy mental conditions. Table 2 shows portions of sample text picked up from posts scoring high on topic 6.

### 5.2.4 Depression, Pandemic and the Student community

Dealing with depression and other mental disorders is in general very hard for the young. The data set we analyzed corroborated that the pandemic added more stress to their otherwise everyday struggle. For many of them, self-isolation led to relapse or surge. Table 4 shows portions from a few samples that reported this issue, captured by topic 9.

## 6 Conclusion

Depression and other mental health issues are persistent with many people, and the social media harbors content where users express their state of mind, discuss problems, and seek counseling. The COVID-19 period presents possibly new factors that stimulate depression (and anxiety) and is the main focus of our study. Specifically, we identified concerns posted by the student community within

Table 2: Table showing typical examples of text corresponding to different categories in Topic 5 (Educational concerns) and Topic 6 (Employment concerns).

| Stress factors identified | Sample Text (Topic 5) |
|---|---|
| Heavier workload than regular classes | "this online school is killing me It's just double the homework, teachers speak to us about the homework for the same time I would usually go to school and then Leave us with double the homework we would usually have." |
| Fear of missing assignments | "Somehow online school just makes the misery worse and I'm not doing me online assignments as I just don't have the energy if that makes sense. Then since I didn't do the assignments I wnd up getting lower grades and that also just makes thing worse." |
| Missing friends in School | "I miss my friends so much, they were pretty much the only reason I kept going to school, and now I don't have that." |
| Locked down with non-cooperative parents | 1."Also everyone gets so disappointed when I fail to turn in some homework in time, and my parents threaten me that I won't be able to pass the class if I'm not 100% successful."
2. "My mom doesn't let me do anything else than work. She constantly reminds of everything I have to do and I don't feel like I have a purpose anymore."
3. "My parents are constantly pressuring me to do well enough to get scholarships. I understand that Scholarships are important but they make me feel like I'm a failure if I'm not a perfect student." |
| | Sample Text (Topic 6) |
| Lost job | 1. "this pandemic is going to kill me i'm a recent college graduate thats been having a hard time finding work in my field. ... my restaurant is still open for take out, but they laid me off. i don't qualify for unemployment because i was only at this job for a month and didn't work my last year of college."
2. "fired and feel a failure the company i ve been working for in the past years is closing (used to work part time because i am a caregiver of my elderly parents)" |
| Isolation woes | "how ironic being isolated, when a lot of us were already in that place. ...in late february i get a job offer from my old 13hr to 45k a year in a new city. ... but then also i have no friends here." |

certain groups in Reddit. The comments clearly reveal that students who have already declared themselves as suffering from mental illnesses like depression are suffering additional stress, both neurotic and psychotic in nature. The topic modeling revealed specific concerns - related to people, negative emotions, family issues, daily activities, education, employment, friends and relationships, feelings, health issues, and even issues of their past life. While we have shared our views on how online education platforms need to accommodate the special needs of the student community as a whole, hopefully, the insights shared here will also be useful for socio-economic planning.

Our future plans are to study the conversations and interaction patterns within these communities. This can help further in understanding the role of positive influences and key users who play those roles within a community. Since social media is an integral part of social lives today, these kinds of insights can help in engaging it more fruitfully towards ensuring community well-being.

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

Table 3: Table showing typical examples of text corresponding to different categories in Topic 3 (Family) and Topic 7 (Friendship).

|  | Sample Text (Topic 3) |
|---|---|
| **Abusive parents** | 1. "i hate my abusive dad i wish i was just born without a father like a immaculate conception... this quarantine is forcing me to be stuck in the the house with my whole family mostly my abusive father."
2. "i just found out my upbringing was abusive and now i want to die. my parents are abusive physically, mentally, and financially. i'm a 23 year old male. i broke mentally over the last of respect for boundaries. ever since then, i have left home and i have been in quarantine with my gf and have time to think over my life."
3. "nightmares about my abusive father so my father has been abusive since i was a little kid. he beat me and verbally abused me everyday."
4. "my mom abused me for years and one day i was self harming at school and she squeezed my arm so hard it bruised and i went a mental hospital" |
|  | Sample Text (Topic 7) |
| **Missing friends in General** | 1. "so quarantine is kinda getting to me. i can't really go out, i'm glad i'm not working anymore. i just miss my friends. some thinks it's ok and they've let me know they still care but to what extent."
2. "ever since quarantine i realised i really dont have friends. nobody texts me and here i was thinking that my friends would help me get through it all."
3. "just a dumb question i have depression, and the whole quarantine thing has made it really come out, but i've been spending a lot of time talking to friends on discord." |

Levine, Blaine Price, Graham Pike, Bashar Nuseibeh, et al. 2020. Altruism and anxiety: Engagement with online community support initiatives (OCSIs) during Covid-19 lockdown in the UK and Ireland. *arXiv preprint arXiv:2006.07153*.

Jerome H Friedman. 2001. Greedy function approximation: a gradient boosting machine. *Annals of statistics*, pages 1189–1232.

Md Akhtarul Islam, Sutapa Dey Barna, Hasin Raihan, Md Nafiul Alam Khan, and Md Tanvir Hossain. 2020. Depression and anxiety among university students during the COVID-19 pandemic in Bangladesh: A web-based cross-sectional survey. *PloS one*, 15(8):e0238162.

Bennett Kleinberg, Isabelle van der Vegt, and Maximilian Mozes. 2020. Measuring emotions in the Covid-19 real world worry dataset. *arXiv preprint arXiv:2004.04225*.

Irene Li, Yixin Li, Tianxiao Li, Sergio Alvarez-Napagao, and Dario Garcia. 2020. What are we depressed about when we talk about covid19: Mental health analysis on tweets using natural language processing. *arXiv preprint arXiv:2004.10899*.

Navonil Majumder, Soujanya Poria, Alexander Gelbukh, and Erik Cambria. 2017. Deep learning-based document modeling for personality detection from text. *IEEE Intelligent Systems*, 32(2):74–79.

Curtis Murray, Lewis Mitchell, Jonathan Tuke, and Mark Mackay. 2020. Symptom extraction from the narratives of personal experiences with COVID-19 on Reddit. *arXiv preprint arXiv:2005.10454*.

James W Pennebaker, Martha E Francis, and Roger J Booth. 2001. Linguistic inquiry and word count: Liwc 2001. *Mahway: Lawrence Erlbaum Associates*, 71(2001):2001.

Petersen, Andrea. 2020. Coronavirus Turmoil Raises Depression Risks in Young Adults. https://www.wsj.com/articles/coronavirus-turmoil-raises-depression-risks-in-young-adults-11597066200. [accessed 28 August 2020].

pushshift.io. 2020. Calling API for dumping dataset. https://pushshift.io/api-parameters. Accessed: 2020-05-01.

Stuart Rose, Dave Engel, Nick Cramer, and Wendy Cowley. 2010. Automatic keyword extraction from individual documents. *Text mining: applications and theory*, 1:1–20.

Yla R Tausczik and James W Pennebaker. 2010. The psychological meaning of words: LIWC and computerized text analysis methods. *Journal of language and social psychology*, 29(1):24–54.

Isabelle van der Vegt and Bennett Kleinberg. 2020. Women worry about family, men about the econ-

Table 4: Samples from posts mentioning "Relapse due to isolation"

| | Sample Text(Topic 9) |
|---|---|
| 1. | "and this past year has been, in contrast, heaven. i am more much functional, and i very much enjoy sanity. however, in light of the covid-19 crisis, i'm thinking that all this self-isolation is getting to me. these past 1-2 weeks, i've been noticing my ability to keep my thinking clear and organized has suffered... and i'm writing this while trying to ignore the thoughts in my head that i cannot control telling me there are people out to get me." |
| 2. | "so today im coming up to my 17th birthday in 9 days which im going to spend in isolation due to quarantine, im alone i have no one but my family to celebrate it but my family struggles to understand my interests as i would consider my self nerdy. no one to be compasionate for no one to hapy for left no one to relate to left no one to share this story with." |
| 3. | "self isolation has brought my depression back after 2 years of not having an episode. need help finding the motivation to do anything. this past week i've really noticed my mental health getting worse by the day." |
| 4. | "i'm feeling very depressed because life has lost all quality amid corona and i'm not even sure i want to be alive anymore. i'm only 20 years old so i don't even have a lot of fond memories to look back on and i feel like i am wasting my youth in isolation." |

omy: Gender differences in emotional responses to COVID-19. *arXiv preprint arXiv:2004.08202*.

World Health Organization. 2020. WHO Director-General's opening remarks at the media briefing on COVID-19 - 11 March 2020. https://www.who.int/dg/speeches/detail/who-director-general-s-opening-remarks-at-the-media-briefing-on-covid-19—11-march-2020. [accessed 5 May 2020].

Jia Xue, Junxiang Chen, Ran Hu, Chen Chen, ChengDa Zheng, and Tingshao Zhu. 2020. Twitter discussions and concerns about COVID-19 pandemic: Twitter data analysis using a machine learning approach. *arXiv preprint arXiv:2005.12830*.

Han Zhang, Paula Nurius, Yasaman Sefidgar, Margaret Morris, Sreenithi Balasubramanian, Jennifer Brown, Anind K Dey, Kevin Kuehn, Eve Riskin, Xuhai Xu, et al. 2020. How Does COVID-19 impact Students with Disabilities/Health Concerns? *arXiv preprint arXiv:2005.05438*.

Jianlong Zhou, Shuiqiao Yang, Chun Xiao, and Fang Chen. 2020. Examination of community sentiment dynamics due to Covid-19 pandemic: a case study from Australia. *arXiv preprint arXiv:2006.12185*.
