# OpenReview forum: "Identifying pandemic-related stress factors from social-media posts - effects on students and young-adults"
_EMNLP/2020/Workshop/NLP-COVID — NLP-COVID19-EMNLP Poster_

### Official Review · AnonReviewer2 · 2020-09-10
**Review of depression-related conversations on Reddit**

**Rating:** 5
**Confidence:** 4

**Review:**

This study aimed to analyze depression-related issues reported by Reddit users using statistical and NLP techniques. Strengths of this study include the methodologies used to systematically analyze content using linguistic techniques to find out the stress-inducing factors. Weaknesses of this study include the lack of supporting documentation suggesting Reddit is a place where those suffering from depression and/or related symptoms express themselves, a lack of justification for choosing Reddit over a traditional survey-based approach, and the failure to rely on previously established terms to identify health concerns.

To improve this paper I’d like to see the authors answer the following questions: 1)Did the authors consider using the Unified Medical Language System (UMLS) to identify depression-related terms? Why or why not? 2) What degree of statistical significance did this study rely on? 3) Why did this study introduce a sentiment analysis in the discussion section? This seemed out of place.

Additionally, this study made a number of claims without appropriate supporting documentation. I’d like to see this corrected, for example, how did this study determine that “the data here spans across geographical, demographic and socio-cultural barriers,” and how did it determine statistical significance? Please explain.

---

> ### Author Response · Authors · 2020-09-30
> **Re: Review of depression-related conversations on Reddit**
>
> We are thankful to reviewer for the valuable feedback. We like to address some of the comments below:
>
> Comment 1): Did the authors consider using the Unified Medical Language System (UMLS) to identify depression-related terms? Why or why not?
>
> Reply: For this study, we have not used UMLS to identify depression-related terms. On Reddit each post is submitted to a particular subreddit which has a particular topic. We have considered only those subreddits which are related to depression.
>
>  Comment 2): What degree of statistical significance did this study rely on?
>
> Reply: This work was conducted as a text mining approach – the objective was to discover the topics of conversation.
>
>  Comment 3): Why did this study introduce a sentiment analysis in the discussion section? This seemed out of place.
>
> Reply: We have included the sentiment analysis to determine the "sense” in which the family and friends are mentioned by the depressed person. In this pandemic situation, a person has been mostly surrounded by family members and cut-off from friends physically. It was found that friends were mentioned in “positive” sense and family members were mentioned in “negative” sense. Though anecdotal at this point, we discovered that family members are usually not “supportive” of a depression patient and they usually find mental support from friends. Being cut-off from friends during the pandemic was an important stress factor during the pandemic.
>
> Comment 4): how did this study determine that “the data here spans across geographical, demographic and socio-cultural barriers,” and how did it determine statistical significance? Please explain.
>
> Reply: This was observed from country name mentions. Due to lack of space the information  was not reported in the paper – but it can be done in the final version.

---

### Official Review · AnonReviewer3 · 2020-09-24
**Data source is limited, the work is not solid enough**

**Rating:** 4
**Confidence:** 4

**Review:**

The study collected posts from Reddit and use common NLP techniques (e.g. topic modelling, sentiment analysis) to find out some stress factors and effects among young people. The findings are interesting and intuitive, but I am concerned about two things: 1) Is the data source limited, why not include other social media like twitter? Is there a fact that young people use Reddit more often than other social media ? 2) The authors put much effort in topic modelling, it would be better to give some qualitative analysis rather than only quantitative insights. 3) Not sure what is the goal of sentiment analysis ? what is your accuracy then ?

---

> ### Author Response · Authors · 2020-09-30
> **Re: Data source is limited, the work is not solid enough**
>
> We are thankful to the reviewers for their valuable feedback. We like to address some of the comments below:
>
> Comment 1): Is the data source limited, why not include other social media like twitter? Is there a fact that young people use Reddit more often than other social media?
>
> Reply: Reddit is a social-discussion platform that is reported as very popular among students. Reddit ranks 17th in most popular websites worldwide. It is structurally and content-wise very rich. Hence was chosen for analysis. There was no need to filter content by keywords as one has to select appropriate tweets.
>
> Comment 2): The authors put much effort in topic modeling, it would be better to give some qualitative analysis rather than only quantitative insights.
>
> Reply: That is our ongoing work with psychological experts
>
> Comment 3): Not sure what is the goal of sentiment analysis ? what is your accuracy then ?
>
> Reply: We have included the sentiment analysis to determine the "sense” in which the family and friends are mentioned by the depressed person. In this pandemic situation, a person has been mostly surrounded by family members and cut-off from friends physically. It was found that friends were mentioned in “positive” sense and family members were mentioned in “negative” sense. Though anecdotal at this point, we discovered that family members are usually not “supportive” of a depression patient and they usually find mental support from friends. Being cut-off from friends during the pandemic was an important stress factor during the pandemic.  We have used the well known VADER sentiment analysis tool for getting sentiments.

---

### Official Review · AnonReviewer4 · 2020-09-27
**Interesting analysis of pandemic-related stress factors with an incomplete evaluation**

**Rating:** 5
**Confidence:** 3

**Review:**

In this work, the authors use a dataset built from reddit posts to identify pandemic-related stress factors. The framing of the problem is nicely outlined, and the focus on college students is a relevant one. The dataset and baseline comparsion dataset are well constructed, but when 'filtering' for only college student related posts it decreases in size to almost being only 5% of the collected set. A discussion of the demographics of reddit users would have better strengthened the authors argument. The techniques applied to the dataset are sufficient to elucidate what the authors are trying to show, but many details are obscured from the reader, hindering both reproducibility, and clarity of the writeup. The sentiment analysis part is a bit dry and lacking in additional details/purpose. Seems rather odd that on Figure 4 there are no neutral posts related to family. One of the major issues is the lack of usage of appropriate vocabularies/lexicons for the depression evaluation, as well as the lack of a detailed qualitative evaluation. Overall, the paper needs additional details to be self-standing and extra clarity and rigor is needed for publication worthiness.

---

> ### Author Response · Authors · 2020-09-30
> **Re: Interesting analysis of pandemic-related stress factors with an incomplete evaluation**
>
> We are thankful to the reviewers for their valuable feedback. We like to address some of the comments below:
>
> Comment 1):  The sentiment analysis part is a bit dry and lacking in additional details/purpose. Seems rather odd that on Figure 4 there are no neutral posts related to family.
>
> Reply: We have included the sentiment analysis to determine the "sense” in which the family and friends are mentioned by the depressed person. In this pandemic situation, a person has been mostly surrounded by family members and cut-off from friends physically. It was found that friends were mentioned in “positive” sense and family members were mentioned in “negative” sense. Though anecdotal at this point, we discovered that family members are usually not “supportive” of a depression patient and they usually find mental support from friends. Being cut-off from friends during the pandemic was an important stress factor during the pandemic. We have used the well known VADER sentiment analysis tool for getting sentiments. The results for sentiments for the "family" category  not include any neutral posts. So we have reported our results as we got.
>
> Comment 2): One of the major issues is the lack of usage of appropriate vocabularies/lexicons for the depression evaluation, as well as the lack of a detailed qualitative evaluation.
>
> Reply: We have used Linguistic Inquiry and Word Count(LIWC) tool that calculates the degree to which various categories of words are used in a text.